# "*Female genital schistosomiasis is a sexually transmitted disease*": Gaps in healthcare workers' knowledge about female genital schistosomiasis in Tanzania

**Humphrey D. Mazigo**[1]*, **Anna Samson**[2], **Valencia J. Lambert**[3], **Agnes L. Kosia**[4], **Deogratias D. Ngoma**[5], **Rachel Murphy**[6], **Dunstan J. Matungwa**[7,8]

1 Department of Parasitology and Entomology, Weill Bugando School of Medicine, Catholic University of Health and Allied Sciences, Mwanza, Tanzania, 2 Department of Behavioral Sciences, School of Public Health, Catholic University of Health and Allied Sciences, Mwanza, Tanzania, 3 Center for Global Health, Weill Cornell Medicine, New York, NY, United States of America, 4 School of Nursing, Catholic University of Health and Allied Sciences, Mwanza, Tanzania, 5 Accelerating the Sustainable Control and Elimination of Neglected Tropical Diseases—Crown Agents, London, United Kingdom, 6 Crown Agents, London, United Kingdom, 7 Department of Sexual and Reproductive Health, National Institute for Medical Research, Mwanza, Tanzania, 8 Department of Anthropology, School of Arts and Sciences, Rutgers University, New Brunswick, NJ, United States of America

* humphreymazigo@gmail.com

**Data Availability Statement:** The qualitative data for this study contains information that can make participants identifiable. In addition, participants did

## Abstract

Female Genital Schistosomiasis is a gynecological disease that is a complication of parasitic *Schistosoma haematobium* infection and affects at least 40 million girls and women, mostly in sub-Saharan Africa. Little is known about how healthcare workers in endemic areas perceive and manage (diagnose and treat) Female Genital Schistosomiasis. We conducted cross-sectional focus group discussions and key informant interviews among healthcare workers in northwestern Tanzania. Healthcare workers, particularly those working in areas where *S. haematobium* is highly endemic, were purposively sampled to participate in the study. Discussions and interviews were digitally recorded, transcribed, and analyzed using NVivo version 12. Most healthcare workers lacked knowledge and skills to manage Female Genital Schistosomiasis. They also had multiple misconceptions about its aetiology, modes of transmission, symptoms, and management. Healthcare workers did not consider Female Genital Schistosomiasis in differential diagnoses of women presenting with gynecologic symptoms except sometimes in patients who did not respond to the initial therapy for sexually transmitted infections. Healthcare facilities had limited capacity to manage Female Genital Schistosomiasis. Our findings show critical gaps in both the knowledge of healthcare workers to manage Female Genital Schistosomiasis and in the capacity of healthcare facilities to manage it. To fill these gaps, two urgent needs must be fulfilled: first, training healthcare workers (particularly those working in schistosomiasis-endemic settings) on Female Genital Schistosomiasis, and second, stocking healthcare facilities with necessary medical equipment and supplies for managing this disease.

not consent to have their full transcripts made publicly available. The data policy exception related to privacy concerns pertains in this case. Data are available on request from the Directorate of Research and Publication of the Catholic University of Health and Allied Sciences (https://www.bugando.ac.tz/) at vc@bugando.ac.tz

**Funding:** This work was supported by the Task Force for Global Health, Coalition for Operational Research on Neglected Tropical Diseases (NTD-SC-208.2D). Humphrey D. Mazigo received additional funding from UK Foreign Commonwealth and Development Office (FCDO) through the Accelerating the Sustainable Control and Elimination of Neglected Tropical Diseases (ASCEND) programme (FCDO grant number PO-8374). The views expressed in this publication are those of the authors and not necessarily those of the funding agencies. Its contents are solely the responsibility of the authors and do not necessarily represent the official views of the supporting offices. The funders had no role in study design, data collection and analysis, decision to publish, or preparation of the manuscript.

**Competing interests:** The authors have declared that no competing interests exist.

## Introduction

Female Genital Schistosomiasis (FGS) is a gynaecological disease caused by *Schistosoma haematobium*, a parasitic worm that is acquired by skin contact with freshwater contaminated by schistosome cerceriae [1, 2]. Communities in which the infection is most endemic have limited access to clean water and healthcare services [2]. Up to 150 million adolescent girls and women are estimated to be at risk of FGS [2, 3] and about 16–56 milion womens are living with FGS, with the majority of these in sub-Saharan Africa [1, 4]. The variability of these estimates points to the fact that this neglected tropical disease is not well studied and frequently not prioritized by local, regional, and global health policy makers [5, 6].

FGS results from the inflammatory responses to eggs of *S. haematobium* that become trapped in the female reproductive tract, where they cause fibrosis and scarring of genital tissues [4]. Clinical symptoms of FGS in women and adolescent girls include abnormal vaginal discharge, abdominal and pelvic pain, dyspareunia, and post-coital and contact bleeding [7–9]. Chronic FGS has been associated with infertility, ectopic pregnancies, abortion, premature birth and low birthweight [4]. Moreover, growing evidence suggests that FGS increases the risk of acquiring human immunodeficiency virus (HIV) [10–14].

Given the overlap of symptoms and screening modalities, there has been a call to integrate the separate healthcare services addressing each of these diseases into broader sexual and reproductive health (SRH) prevention services [6]. However, to date, in many FGS endemic countries, care for FGS is still separated from both HIV and HPV/cervical cancer prevention [6] and health systems in most FGS endemic countries lack the capacity to identify and diagnose women and girls at risk of FGS infection and to manage FGS cases [3].

In order to support the integration of FGS, HIV, and HPV interventions into a comprehensive SRH package, there is a critical need to understand existing gaps in healthcare workers' ability to manage FGS and to determine availability of essential infrastructure within the primary healthcare system. Moreover, it is essential to understand potential challenges that may be faced by healthcareworkers in providing comprehensive SRH care that incorporates FGS diagnosis and treatment. Therefore, this study aimed to understand healthcare workers' knowledge, attitudes, and perceptions (KAP) on FGS and to determine gaps in health facilities' capacity for managing FGS cases in northwestern Tanzania.

## Methods

### Study setting

This study was conducted in Itilima, Maswa, Misungwi, Shinyanga, and Kwimba districts in northwestern Tanzania. Data from the National Neglected Tropical Diseases Program indicate that *S. haematobium* is highly endemic in these districts [15]. The prevalence of *S. haematobium* differs from one village to another across these districts, with some villages experiencing high levels of transmission of this parasitic worm [16, 17]. Data from the National Strategic Plan for controlling schistosomiasis indicate that the overall prevalence of *S. haematobium* in these districts is >50%, but varies from one village to another [15, 18].

The districts have conducive environmental conditions for the transmission of *S. haematobium*. Overall, the districts' mean annual rainfall ranges between 930–1,200mm and the ambient temperature ranges between 25˚C and 28˚C. The maximum temperature is experienced between August and October, the period with low transmission of *S. haematobium*. The districts have permanent and seasonal freshwater rivers, marshes, swamps, and ponds which create a good breeding environment for freshwater snails, the intermediate hosts of *Schistosoma* [17]. Rice farming, an activity that oftentimes exposes the farmer's skin to cercariae-infested

water, is one of the main economic activities in these districts. Although one cross-sectional study has previously reported that FGS is a public health problem among women and adolescent girls in these districts, with prevalence ranging from 0 to 10%, the disease remains understudied [11]. In these districts, mass drug administration (MDA) using praziquantel is the main control approach and targets only school-age children.

## Study design and recruitment of participants

This study used Focus Group Discussions (FGDs) (see S1 File) [19–21] and Key Informant Interviews (KIIs) (see S2 File) [22] to collect cross-sectional qualitative data from purposively selected participants. All study participants—male and female—working in different healthcare facilities across all five study districts were eligible to participate. Working with the chairpersons and executive officers of the villages where the study was conducted, the study team approached 39 healthcare facilities and purposively selected participants for FGDs and KIIs. The healthcare worker qualified to participate in the FGD if they provided a written informed consent including the consent to audio-record their views aired during the FGD. Using these criteria, we selected 53 healthcare workers from different healthcare facilities (Table 1) to participate in 9 FGDs.

The healthcare worker qualified to be a key informant if they held a leadership position at the healthcare facility: either overseeing the whole facility or a department/unit. In addition, they were eligible to participate if they provided a written informed consent including the consent to audio-record the interview. In total, we conducted 31 KIIs in all five study districts involving 31 participants (13 male and 18 females) (Table 2).

## Data collection and sample size

Data collection was conducted in September 2020. The principal investigator recruited supervisors and research assistants with experience in conducting qualitative research. They were then given a refresher training on using FGDs and KIIs to collect qualitative data. The study team pre-tested the semi-structured topic guides before going for actual fieldwork. All FGDs and KIIs were conducted at a selected healthcare facility after working hours to avoid interruptions and maintain privacy. Led by research assistants, all FGDs and KIIs were conducted in Kiswahili and audio-recorded using a digital audio-recorder after the participants gave a written consent. Every research assistant wrote notes on how the FGD or KII had gone and presented them at a debriefing meeting at the end of every data collection day.

**Table 1. Study participants, FGDs conducted, and study districts.**

| FGD code | Participants | Participants' gender | | District |
|---|---|---|---|---|
| | | Male | Female | |
| 01 | Healthcare workers | 3 | 2 | Itilima |
| 02 | Healthcare workers | 3 | 4 | |
| 03 | Healthcare workers | 2 | 3 | |
| 04 | Healthcare workers | 2 | 5 | Misungwi |
| 05 | Healthcare workers | 3 | 3 | |
| 06 | Healthcare workers | 5 | 0 | Kwimba |
| 07 | Healthcare workers | 5 | 0 | |
| 08 | Healthcare workers | 4 | 2 | Shinyanga Rural |
| 09 | Healthcare workers | 1 | 6 | Maswa |
| | | 28 | 25 | |

**Table 2. Key informants, number of KIIs, and study districts.**

| KII code | Participants | Participants' gender | | District |
|---|---|---|---|---|
| | | Male | Female | |
| 01 | Registered Nurse | M | | Itilima |
| 02 | Laboratory Technician | M | | |
| 03 | Laboratory Technician | | F | |
| 04 | Nurse | | F | |
| 05 | Nurse | | F | |
| 06 | Enrolled Nurse | | F | |
| 07 | Medical Doctor In-Charge | M | | |
| 08 | Laboratory Assistant | | F | |
| 09 | Midwife, Acting In-Charge | | F | |
| 10 | Clinical Officer In-Charge | M | | |
| 11 | Clinical Officer Midwife | | F | |
| 12 | Medical Attendant | | F | |
| 13 | Clinical Officer In-Charge | M | | |
| 14 | Laboratory Assistant | | F | |
| 15 | Midwife In-Charge | M | | |
| 16 | Nurse | | F | Misungwi |
| 17 | Laboratory Technician | M | | |
| 18 | Clinical Officer | M | | |
| 19 | Nurse | M | | |
| 20 | Nurse | | F | Kwimba |
| 21 | Clinical Officer In-Charge | M | | |
| 22 | Nurse Midwife | | F | |
| 23 | Laboratory Technician | M | | |
| 24 | Laboratory Technician | M | | |
| 25 | Laboratory Technician | M | | Shinyanga Rural |
| 26 | Nurse | | F | |
| 27 | Clinical Officer In-Charge | | F | |
| 28 | Clinical Officer Assistant In-Charge | | F | Maswa |
| 29 | Nurse | | F | |
| 30 | Nurse | | F | |
| 31 | Laboratory Technician | | F | |
| | Total | 13 | 18 | |

We determined our sample size of 9 FGDs and 31 KIIs through attaining saturation. However, rather than attaining saturation through finding the point where no new information or themes are generated [23–25], we attained saturation by obtaining the most salient items and themes that were pre-determined in FGD and KII semi-structured topic guides [26]. Our salient items and themes on urogenital schistosomiasis were awareness, perceived prevalence, symptoms, aetiology, modes of transmission (including misconceptions), groups of people at risk of infection, and body parts affected by urogenital schistosomiasis. Our salient items and themes on FGS were awareness, aetiology and modes of transmission, symptoms, associating FGS with other diseases or medical conditions, community perception of women and girls suffering from FGS, treatment seeking behavior for FGS, healthcare facilities' capacity to manage FGS, challenges facing healthcare workers and related to FGS, and interventions to prevent and control the transmission of FGS.

## Data processing and analysis

After completing fieldwork activities, all audio-recorded FGDs and KIIs were transcribed verbatim. The second and third authors sampled a few transcripts to validate if the transcript was a representation of the audio. Data were coded using both concept-driven (deductive) and data-driven (inductive) coding approaches [27]. Drawing on the FGD and KII semi-structured topic guides, second, third, fourth, and the last authors prepared an initial codebook with deductive codes. Thereafter, using data-driven approach, the second, third, and fourth authors coded 2 transcripts each (1 FGD and 1 KII), identified additional codes and themes, and redefined some of the initial codes to produce a final version of the codebook. All transcripts were then imported on the NVivo 12 Plus software. Applying the data-driven approach, the second and third authors coded all the transcripts using the final version of the codebook. Finally, we grouped all the themes into specific categories and selected representative quotes (and translated then into English) to illustrate the study findings.

## Ethical considerations

Ethical approvals to conduct this study were obtained from the Lake Zone Institutional Review Board (certificate number MR/53/100/649) and Weill Cornell College of Medicine (certificate number 20–07022381). The study received further permission from the authorities in the regions and districts where the study was conducted. All study participants were adults (aged 18 years and above) and each provided a written informed consent. Prior to data collection, sensitization meetings were held with responsible authorities in the study villages and healthcare facilities to create awareness of the study and its data collection methods. Anonymity and confidentiality were maintained throughout the study. However, since the FGD's group setting does not guarantee complete confidentiality as individuals may disclose the content of the discussion to non-study participants after the study, the study team pleaded with the participants to maintain privacy of the content of the discussion and the identities of the participants [21]. No identifiers, particularly personal names and names of healthcare facilities, were used during data collection. Each participant was identified using a unique code number. In this paper, we have also omitted names of healthcare facilities to anonymize the facilities where study participants were working as this information may be used to identify them.

## Results

Our study results are divided into two main sections: the first section focuses on urogenital schistosomiasis while the second section focuses on FGS. Our analysis on urogenital schistosomiasis had seven themes: awareness, perceived prevalence, symptoms, aetiology, modes of transmission (including misconceptions), groups of people at risk of infection, and body parts affected by urogenital schistosomiasis. Regarding FGS, our analysis presents nine closely related themes: awareness, aetiology and modes of transmission, symptoms, associating FGS with other diseases or medical conditions, community perception of women and girls suffering from FGS, treatment seeking behavior for FGS, healthcare facilities' capacity to manage FGS, challenges facing healthcare workers and related to FGS, and interventions to prevent and control the transmission of FGS.

### Healthcare workers' knowledge and awareness about urogenital schistosomiasis

Overall, healthcare workers were aware of urogenital schistosomiasis and its symptoms. Participants mentioned blood in urine as the most common symptom of urogenital schistosomiasis.

The symptom was commonly observed in children. The disease is locally called "*Kisambale, gisambale,* or *kunyola*." However, most participants confused the cause of urogenital schistosomiasis (the etiological agent) with the mode of transmission (contact with infested water in fresh sources such as lakes, rivers, dams, marshes, and swamps) (Table 3). A few participants correctly reported that schistosomiasis was caused by parasitic flatworms called *Schistosoma*. Some participants also held misconcneptions on the transmission of schistosomiasis claiming that it could be transmitted through the fecal-oral route, sexual intercourse, sharing of underwear, and stepping on an infected person's urine.

Participants reported that urogenital schistosomiasis was highly prevalent in the communities they served but that the prevalence had recently declined due to annual mass preventive chemotherapy offered to school aged children. Children aged 3–13 years, adolescent girls and women of child bearing age were noted as the most groups at-risk due to high water contacts relating domestic activities, recreational and economic activities such as paddy farming in swamps. Similarly, men were also considered to be at risk of being infected by schistosomiasis because of their involvement in farming activities.

The most common symptoms related to urogenital schistosomiasis were stomach pain, blood in urine, frequent urination, urinating a small quantity of urine and pain during

**Table 3. Healthcare workers' knowledge and perceptions about urogenital schistosomiasis in northwestern Tanzania.**

| Themes | Summary of the participants' views | Illustrative Quotations |
|---|---|---|
| Awareness of urogenital schistosomiasis | Most participants were aware of urogenital schistosomiasis. Their main sources of information were school and their job. | • "*I have heard about schistosomiasis at least two times: at school, where we learned about it in our classes; and at work, where I have participated in government interventions to implement MDA programs.*" (FGD 08, Healthcare workers, Shinyanga Rural).<br>• "*We were taught about schistosomiasis in primary school, secondary school, and college. And now [we are taught about it] at work.*" (KII 08, Laboratory Assistant, Itilima).<br>• "*You can get information about schistosomiasis from different sources. You can learn this in school: at the university or other colleges. You may read that information as advertisements in the newspapers. You may also see it television or hear it from radio advertisements. We [healthcare workers] deal with schistosomiasis cases at work.*" (FGD 04, Healthcare workers, Misungwi). |
| Perceived prevalence of urogenital schistosomiasis | Most participants perceived urogenital schistosomiasis is perceived as a public health problem but thought that its prevalence has declined due to the delivery of preventive chemotherapy to school children through MDA programs. | • "*Schistosomiasis cases are rare these days. We see them once in a while especially among children during the rainy season. During this season, children swim in seasonal rivers. The number of these rivers increases during the rainy season. Children like to swim in these seasonal rivers a lot. For adults, you may get one or two cases.*" (KII 05, Nurse, Itilima).<br>• "*There are a fewer cases of schistosomiasis in our community these days. They were many in the past because there were no any interventions. But these days, people understand how urogenital schistosomiasis is transmitted and how it can be avoided. For instance, people who go for rice farming activities wear protective gear to prevent the infection.*" (FGD 04, Healthcare workers, Misungwi).<br>• "*Schistosomiasis is not a common problem [these days]. If I prepare a checklist of the top ten diseases [affecting people in this place], it will not be part of it. For instance, [after a very long time], today I had a patient who reported that they might be infected with schistosomiasis.*" (FGD 03, Healthcare workers, Itilima). |
| Symptoms of urogenital schistosomiasis | Most participants were aware of the symptoms of urogenital schistosomiasis including blood in urine (haematuria), abdominal/pelvic pain, difficulty passing urine, pain during urination, and frequent urination. | • "*The main symptom [of urogenital schistosomiasis] is blood in urine.*" (KII 07, Medical Doctor, Itilima).<br>• "*One main symptom is to feel pain in the abdomen, joints, or muscles.*" (FGD 09, Maswa).<br>• "*If you have difficulty passing urine, you feel pain during urination, you experience frequent urination; you might be infected urogenital with schistosomiasis.*" (KII 06, clinical officer, Maswa). |
| Aetiology of urogenital schistosomiasis | Most participants confused the cause and the mode of transmission of schistosomiasis. Some participants regarded swimming in contaminated water as the cause rather than the mode of transmission of schistosomiasis. | • "*Using unsafe water causes schistosomiasis. For instance, bathing in or drinking water that is not boiled. People use water that is not boiled and others swim in stagnant water.*" (FGD 01, Healthcare workers, Itilima). |

(*Continued*)

**Table 3.** (Continued)

| Themes | Summary of the participants' views | Illustrative Quotations |
|---|---|---|
| Modes of transmission of urogenital schistosomiasis | Most participants understood that schistosomiasis is transmitted when a person's skin comes into contact with infested freshwater. Skin contact with contaminated freshwater can occur when people engage in agricultural activities such as paddy farming in the wetlands. | • "*Another agricultural activity in this locality is rice farming which takes place in wetlands with stagnant water. If a person goes to the wetland for rice farming activities after snails have released larvae in the same site, they will then be infected with schistosomes through the skin.*" (FGD 01, Healthcare workers, Itilima). |
| | Skin contact with infested freshwater can occur when engaging in recreational activities such as swimming or bathing in infested water. | • "*Bathing in contaminated pond water contributes to the transmission of schistosomiasis.*" (FGD 03, Healthcare workers, Itilima).<br>• "*You get [infected with] those worms or Schistosoma in stagnant water when you step or swim in it. These parasites penetrate the skin and spread to the urinary system.*" (FGD 01, Healthcare workers, Itilima). |
| | A few participants understood that the transmission of schistosomiasis occurs when people suffering from schistosomiasis contaminate freshwater sources with their excreta containing parasite eggs (which then hatch in water). | • "*Some people bath and urinate in the water. [These practices] facilitate the transmission of urogenital schistosomiasis from an [infected] person who urinates in the water to the [uninfected] person who is taking a bath.*" (FGD 02, Healthcare workers, Itilima).<br>• "*An infected person can transmit urogenital schistosomiasis if they urinate or defecate in water sources. Urogenital schistosomiasis is also transmitted through open defecation. When it rains in places with feaces, surface runoff carries the feaces down the slope where it contaminates the freshwater source.*" (KII 01, Registered Nurse, Itilima). |
| | Most participants had misconceptions on the transmission of urogenital schistosomiasis. For instance, some participants believed that schistosomiasis can be transmitted through sexual intercourse. | • "*Urogenital schistosomiasis can also be transmitted through sexual intercourse. If one partner has schistosomiasis, it is possible to transmit it to their uninfected partner during sexual intercourse.*" (FGD 02, Healthcare workers, Itilima). |
| | Some participants believed that schistosomiasis can be transmitted through the feacal-oral route, that is, through drinking contaminated or unboiled freshwater. | • "*Schistosomiasis is transmitted [. . .] through the feacal-oral route. When a person with schistosomiasis urinates or defecates in a freshwater source, larvae are released into the water. When an uninfected person drinks this contaminated water, they get infected with schistosomes.*" (KII 01, Registered Nurse, Itilima).<br>• "*If a person is suffering from urogenital schistosomiasis and they defecate in water, they can infect others. For instance in this case, if another person drinks this water [that is already contaminated with faeces], they will be infected with schistosomes.*" (FGD 05, Healthcare workers, Misungwi). |
| Groups of people at risk of urogenital schistosomiasis infection | Most participants understood that all groups of people are at risk of urogenital schistosomiasis infection but their levels of risk differ. Children, both males and females, are at risk of urogenital schistosomiasis infection because of engaging in activities that involve skin contact with contaminated freshwater such as swimming in rivers and ponds. | • "*Children, especially boys aged 7 to 14 years, are at risk of being infected with schistosomes. During the rainy season, you will find children aged between 7 and 14 swimming in the ponds. That is where they get infected [with schistosomes].*" (FGD 06, Healthcare workers, Kwimba).<br>• "*Children are affected like other groups people. For instance, if a person with urogenital schistosomiasis urinates in the freshwater source and leaves the parasites there, children will be infected if they go to swim in this freshwater source.*" (KII 06, Enrolled Nurse, Itilima).<br>• "*Children are fond of playing in the risk environment [without knowing the risk]. They may play in dirty water thinking that this right [but they are actually exposing themselves to infections]. If you forbid them from doing such things [risk activities], it is like you have told them to do it more.*" (KII 15, Nurse Midwife, Itilima). |
| | Women, particularly women of reproductive age, are at risk of urogenital schistosomiasis infection because of performing activities—such as fetching water for domestic use and farming—that involve skin contact with contaminated freshwater | • "*Women aged above seventeen or eighteen years are mostly affected by schistosomiasis. This is because they do most domestic activities including fetching water.*" (FGD 02, Healthcare workers, Itilima).<br>• "*The nature of activities that women do expose them to schistosomiasis. For example, women do not have farm boots to protect themselves from having skin contact with infested freshwater when they go for rice farming activities in wet areas.*" (KII 08, Laboratory Assistant, Itilima).<br>• "*Women of childbearing are at more risk of being infected with schistosomes because of the anatomy of their reproductive system.*" (FGD 03, Healthcare workers, Itilima). |
| | Men are at risk of urogenital schistosomiasis infection because they also get involved in activities that involve skin contact with contaminated freshwater such as paddy farming. | • "*Because men also take part in rice farming activities, they end up having skin contact with infested freshwater that exposes them to schistosomes.*" (KII 02, Laboratory Technician, Itilima). |
| Body parts affected by urogenital schistosomiasis | Most participants understood most of the body parts affected by schistosomiasis. Most participants mentioned organs of both the urinary system and gastrointestinal system. | • "*Schistosomiasis mainly affects the abdomen, the reproductive system including the genitals and may be the spinal cord.*" (KII 08, Laboratory Assistant, Itilima).<br>• "*Schistosomiasis mainly affects the urinary system because this is the area where eggs are deposited.*" (KII 02, Laboratory Technician, Itilima).<br>• "*Schistosomiasis affects two main areas: the urinary system and gastrointestinal system. This is to say there is urogenital schistosomiasis and intestinal schistosomiasis.*" (KII 09, Nurse Midwife, Itilima). |

urination. A few participants also mentioned fever and headache. The urinary bladder was noted to be the common body part affected by urogenital schistosomiasis. Other parts were male and female reproductive organs, legs, stomach and cervix, spinal cord, waist, liver and kidneys. Table 3 summarizes healthcare workers' responses regarding knowledge and perceptions about urogenital schistosomiasis in northwestern Tanzania.

## Healthcare workers' awareness of Female Genital Schistosomiasis

Most participants reported that they had never heard about FGS. They heard about it for the first time in this study. In addition, most participants were unaware that schistosomiasis can affect the male and female reproductive systems. But they knew about urogenital schistosomiasis which most understood as affecting the bladder, among other organs.

"*FGS? I have never heard about it. I know about urogenital schistosomiasis but not FGS.*" (FGD 08, Healthcare workers, Shinyanga Rural).

"*I have heard of it for the first time today when you were explaining the objectives of the study. I then read about it in the consent form. I had never heard or read about it before.*" (KII 09, Nurse Midwife, Itilima).

"*I know that schistosomiasis can affect the bladder. But I did not know that those parasitic worms can also affect the female reproductive system.*" (FGD 01, Healthcare workers, Itilima*).*

A few healthcare workers reported that they had heard about FGS during their college training and at their working stations (healthcare facilities) particularly when they were involved in the national schistosomiasis treatment campaigns. A few others reported that they usually considered FGS as part of the differential diagnosis for diseases affecting the reproductive system but the challenge was getting confirmatory diagnosis.

"*I usually consider FGS when I conduct a medical examination, since I learned about it in college. The challenge is how to confirm it. I fail to diagnose and say with certainty that this is FGS or Pelvic inflammatory disease (PID) because they have similar symptoms.*" (KII 10, Clinical Officer, Itilima).

"*I treat FGS as a differential diagnosis of Pelvic inflammatory disease (PID). Usually, when I prescribe the PID treatment and the patient does not respond well, I change and give them praziquantel. Then, they respond very well.*" (KII 13, Clinical Officer, Itilima).

## Aetiology and transmission of Female Genital Schistosomiasis

Most participants did not know the aetiology of FGS. A few participants who explained the aetiology of FGS were those who had learned about it in college.

"*FGS is caused by a bacterium, if I am not mistaken. It is called Schistosoma haematobium or japonicum, something like that!*" (FGD 04, Healthcare workers, Misungwi).

Most participants confused the aetiology and mode of transmission of FGS. Thus, when asked about the aetiology, they explained how they thought FGS was transmitted instead. Many misconceptions were noted and most participants reported that they did not know how FGS is transmitted. A few participants who attempted to explain the transmission of FGS held

misconceptions that it can be transmitted through sexual intercourse or contact with (exposure to) body fluids such as urine or vaginal discharge from a person suffering from the disease.

> "*She can be infected via sexual intercourse, if they [sexual partners] do not use Salama condom.*" (KII 12, Medical Attendant, Itilima).

> "*FGS can be transmitted through sexual intercourse. If the woman has sex with her infected husband, she will also be infected.*" (KII 16, Nurse, Misungwi).

> "*Worms [that transmit FGS] can penetrate an unbroken skin. They can penetrate any part. If a man is suffering from schistosomiasis and has unprotected sex with an uninfected woman, these worms can move from the man to the woman.*" (KII 14, Laboratory Assistant, Itilima).

However, participants who understood the life cycle of schistosomiasis disagreed with the assertion that FGS can be transmitted through sexual contact pointing out that schistosomes need an intermediate host to become infective.

> "*FGS is not directly transmitted until it follows a certain lifecycle. I disagree with those saying that it can be sexually transmitted.*" (FGD 01, Healthcare workers, Itilima).

> "*I also disagree. Schistosomiasis is not transmitted through sexual contact because schistosomes need an intermediate host before they can infect a person.*" (FGD 01, Healthcare workers, Itilima).

### Symptoms of Female Genital Schistosomiasis

Most participants did not know the symptoms of FGS. However, some used their knowledge of urogenital and intestinal schistosomiasis to describe and explain the symptoms of FGS. Those with knowledge of FGS mentioned some of its symptoms including vaginal discharge, abdominal and pelvic pain, pain or difficulty when urinating (dysuria), frequent urination (polyuria), and inflammation of the cervix, endometrium, and/or of the fallopian tubes.

> "*If a woman has FGS, the first symptom will be vaginal discharge. Sometimes the discharge may have a foul smell. She will experience severe lower abdominal pain and back pain.*" (FGD 02, Healthcare workers, Itilima).

> "*A patient might present with lower abdominal pain. Sometimes she can present with per vaginum (PV) bleeding or irregular menstruation.*" (FGD 08, Healthcare workers, Shinyanga Rural).

> "*The patient feels difficulty and pain when urinating. They urinate frequently. They also experience inflammation in different parts of their reproductive system.*" (FGD 09, Healthcare workers, Maswa).

Other participants, however, reported that they had always considered certain symptoms of FGS as signs of other infections (other than FGS). Others argued that what other participants had mentioned as symptoms of FGS could not be used to rule out the possibility of a patient having other diseases or infections.

> "*I have never associated vaginal discharge with schistosomiasis. I always think it a symptom of UTI (Urinary Tract Infection) or any other sexually transmitted infection.* (FGD 02, Healthcare workers, Itilima).

"*I do not think vaginal discharge can confirm [that a patient has] schistosomiasis. It could also be a symptom of a sexually transmitted infection.*" (FGD 08, Healthcare workers, Itilima).

## Associating Female Genital Schistosomiasis with other diseases or medical conditions

Most respondents reported that they did not know how FGS could create or increase the risk of or was associated with other diseases or medical conditions particularly those associated with the reproductive system such as ectopic pregnancy, infertility, miscarriage, and cervical cancer. Most participants also reported that when dealing with patients who presented with symptoms of the aforementioned diseases or medical conditions, they did not consider schistosomiasis in the differential diagnosis. However, a few participants explained how FGS was associated with HIV/AIDS and other STIs.

"*We know that HIV spreads through bodily fluids or bruises. If she engages in sexual intercourse with a man and she has bruises, they can easily get HIV or other sexually transmitted infections.*" (KII 14, Laboratory Assistant, Itilima).

## Community perception of women and girls suffering from Female Genital Schistosomiasis

Participants were asked to reflect on how the communities they serve perceive women and girls who are suffering from FGS. Most participants reported that FGS is considered as an STI and that women and girls with this disease are considered as "prostitutes", having multiple sexual partners, and unfaithful in their intimate relationships. If a married woman is suffering from FGS, community members perceive them as promiscuous and unfaithful to their spouses.

"*The perception of the community is that the person suffering from FGS is involved in multiple sexual relationships. This is what most of the men in this community believe. They will ask [a woman or a girl], 'how did you get this disease? You certainly have multiple sexual partners who have the diseases [sexually transmitted diseases].'*" (KII 12, Medical Attendant, Itilima).

"*Most people especially men think that this disease is sexually transmitted. If a man finds out that his wife has these symptoms [of FGS], he will think she is a prostitute. If he is not educated and advised about these symptoms, they will have a fierce fight.*" (KII 19, Nurse, Misungwi).

When an adolescent girl is suffering from FGS, community members think that she had early sexual debut and perceive her as promiscuous and ill-mannered.

"*The community regards girls with FGS as undisciplined, deviants, and prostitutes. They think FGS is a sexually transmitted disease. This makes it difficult for girls to come for healthcare at the healthcare facilities because they fear being stigmatized by the community members.*" (FGD 01, Healthcare workers, Itilima).

However, when an older woman—considered less sexually active—is suffering from FGS, community members will not stigmatize her. They will regard her symptoms and disease as normal. But other people may think that she has been bewitched or that she engages in unprotected sex.

"*I think the community will have varied views. Some people will regard this as a normal disease because of her old age. Others will say she is undisciplined. She is old but she likes having [unprotected] sex. Others will say she has done something bad to a person, so she has been bewitched. If she is still in her reproductive age, others will think she has had an unsafe abortion. People will not think that she is suffering from FGS.*" (KII 14, Laboratory Assistant, Itilima).

### Treatment seeking behavior for Female Genital Schistosomiasis

Participants reported that most people depend on traditional healers and self-treatment (using natural herbs) to treat FGS. People also purchase drugs from the retail drug shops for self-treatment. When these treatment options fail, they then go to the healthcare facilities. Thus, most patients present to the healthcare facilities late and in most cases when their conditions have worsened.

"*Most patients will first seek treatment from the traditional healers, thinking that they have been bewitched. They will use herbs. When they do not improve, they come to the hospital. But there are a few patients who will think of coming to hospital first [before going for other treatment options].*" (KII 11, Clinical officer, Itilima).

"*The challenge is that [when patients have symptoms of FGS], they start treating themselves using herbs. When they do not feel better, they then go to the hospital.*" (KII 18, Clinical Officer, Misungwi).

"*[When patients have the symptoms of FGS], they go to the retail drug shop and buy drugs for self-medication.*" (KII 29, Nurse, Maswa).

### Healthcare facilities' capacity to manage Female Genital Schistosomiasis

All participants agreed and reported that their healthcare facilities lack FGS diagnostic and treatment capacity. Participants reported that Praziquantel was sometimes not available in the health facilities. Other equipment that were not available in the healthcare facilities were miscroscopes and gynecologic specula. Most healthcare facilities were also under-staffed making it difficult for the available, small number of healthcare workers to manage the number of patients presenting to their facilities.

"*There are no medical equipment [to diagnose FGS]. There is also need to increase the amount of medications [Praziquantel]. However, I think it is because our reports show a small number of cases of schistosomiasis. For instance, if you report every month [. . .] that you only had 2 cases, the government will not think of bringing medical equipment or other supplies [for schistosomiasis].*" (FGD 06, Healthcare workers, Kwimba).

"*There are no enough healthcare workers. For example, we are only three (3) at our facility: the doctor, the nurse, and the medical attendant. We do not have a laboratory technician. But even when we are four (4), our number will still be very small compared to the number of patients we serve.*" (KII 30, Nurse, Maswa).

### Challenges facing healthcare workers and related to Female Genital Schistosomiasis

Participants agreed that most of them lacked knowledge of FGS. This makes it difficult for them to diagnose or consider FGS in differential diagnosis. It also makes it difficult for them to educate their patients on the symptoms of the disease, its modes of transmission, and what

they must or must not do to avoid catching it. These challenges are further compounded by the lack of FGS diagnostic equipment (such as miscroscopes and gynecologic specula) and other supplies including the availability and accessibility of Praziquantel, and the small number of healthcare workers at the healthcare facilities.

"*Lack of knowledge about FGS [is the key challenge] not just in the community, but also to us, the healthcare workers. We do not know about this disease, really.*" (FGD 08, Healthcare workers, Itilima).

### Interventions to prevent and control Female Genital Schistosomiasis

We asked our study participants to suggest the interventions that could be implemented to prevent and ultimately control FGS. We have categorized these interventions in three broad categories as interventions focusing on 1) raising awareness about FGS, 2) strengthening heathcare facilities' capacity to manage FGS cases, and 3) other crucial interventions that were suggested by fewer participants.

### Raising awareness of Female Genital Schistosomiasis

Since most of the participants did not have knowledge of FGS, they suggested that one of the key interventions is to raise awareness of this disease among healthcare workers. Such intervention would be delivered through in-service trainings focusing on schistosomiasis in general and FGS in particular. These trainings will equip healthcare workers with knowledge of the symptoms and complications, aetiology, modes of transmission of FGS, as well as ways to prevent it such as avoiding swimming or wading in contaminated freshwater.

"*I think the government [and other stakeholders] should educate healthcare workers about FGS. If the healthcare worker has little or no knowledge [about FGS], it becomes very difficult for them to help the patients. For instance, since they do not understand the symptoms [of FGS], they will misdiagnose [the disease], and they will thus not give proper medical advice to the patients.*" (KII 05, Nurse, Itilima).

Equipped with knowledge of FGS, healthcare workers could then serve as "ambassadors" and deliver this knowledge to communities they serve. In this way, they could raise awareness of FGS among people and mitigate stigmatization that women and girls face when they are suffering from this diseases or any other STIs.

"*If we [healthcare workers] will be educated about FGS, we will serve as ambassadors and educate the communities where we work. If we educate community members, they will not stigmatize women and girls who have this disease. So, when they have this disease, they will be motivated to come to seek for [medical] advice and care at the healthcare facilities.*" (KII 14, Laboratory Assistant, Itilima).

Participants also suggested that there should be interventions to raise awareness of FGS among community members. They suggested using both community-based and healthcare facility-based education interventions to improve people's knowledge of FGS and address stigma associated with it. Both forms of interventions can improve people's knowledge of the symptoms and complications, aetiology, and modes of transmission of FGS. They can also improve people's understanding of how to prevent oneself from schistosome infection and

help in addressing people's stigmatizing attitudes. Community-based education interventions could be delivered through mass campaigns, household-level training, as well as in collaboration with existing community structures such as schools and religious institutions. Healthcare facility-based education intervention could target women and their children during their pre and postnatal care visits. Both of these interventions could improve people's health seeking behavior by choosing to visit healthcare facilities first instead of other treatment options such as self-treatment and traditional healers.

> "*People in the community need to be educated about FGS. People can be educated in the public settings or in their households. Education sessions can also be offered at the healthcare facilities. These can be delivered to women who come for pre and postnatal care visits every month. These women will be educated about the symptoms of FGS and be encouraged to go to the healthcare facility for diagnosis whenever they have the symptoms [of FGS]. They can also be encouraged to go for diagnosis even when they have not seen any symptoms. This is a good practice.*" (KII 20, Nurse, Kwimba).

> "*People in the community should be educated about schistosomiasis so that they can understand what it is and its symptoms. They should also be educated that whenever they see any of those symptoms [of FGS], they should go to the healthcare facilities for diagnosis. If this will be done, there will be no stigmatization among them and in the families. They will start regarding it as normal disease. This information and where to access healthcare services could be provided in schools, through religious institutions, and in the community [in villages and hamlets].*" (KII 05, Nurse, Itilima).

## Strengthening healthcare facilities to manage Female Genital Schistosomiasis cases

In addition to training healthcare workers on FGS, participants suggested creating a friendly environment at the healthcare facilities as a crucial intervention. They suggested that a friendly environment can be created through training healthcare workers to address their stigmatizing attitudes on women and adolescent girls' sexual and reproductive health (SRH) concerns and gynecological diseases such as FGS. Since stigma undermines people's access to healthcare services, particularly diagnosis and treatment; addressing it is fundamental to delivering quality healthcare and achieving optimal health.

> "*There is a need for creating a friendly environment at the healthcare facilities. If the environment is friendly, people will come to utilize healthcare services. If healthcare workers have stigmatizing attitudes towards their patients, people will fear coming to the heathcare facility [for care and other health services].*" (KII 01, Registered Nurse, Itilima).

They further suggested that a friendly environment at the healthcare facilities could also be created through training healthcare workers on the importance of anonymity of the patients' identities and confidentiality of all the information on patients' illness and treatment.

> "*Healthcare workers must assure their clients that the information about their illness and treatment will be treated with confidentiality. They should also assure them privacy. If you assure the patient that you will not reveal their identities in the community [or elsewhere] that so and so came at the health facility with this or that illness, they will feel comfortable and explain their illness openly. Then you will help them because they will explain their illness in detail.*" (FGD 02, Healthcare workers, Itilima).

The second form of intervention to strengthen the healthcare facilities capacity to manage FGS and other STIs is that the government needs to address the staff shortage by increasing the number of staff (of different cadres) at the healthcare facilities. Through collaboration with other stakeholders, the government should also stock the healthcare facilities with equipment and supplies for FGS (e.g. miscroscopes and gynecologic specula) and other SRH concerns including medication, particularly Praziquantel.

"*Medical equipment and supplies [for diagnosing FGS] should be made available. Healthcare workers should be well trained [on how to manage FGS]. There should be enough healthcare workers. Enough medication [Praziquantel] should be made available.*" (KII 27, Clinical Officer, Shinyanga Rural).

## Other important interventions to prevent and control Female Genital Schistosomiasis

A few participants suggested three other important interventions to prevent and control FGS. First, they suggested that the government and other stakeholders should ensure that every village has enough supply of safe and clean water to prevent the transmission of FGS. Second, they suggested that people should be educated on the importance of using protective gear when they engage in activities that involve skin contact with contaminated freshwater in order to protect themselves from being infected with schistosomes.

"*The government and other stakeholders should make sure that every village gets clean and safe water. The community should also be educated on the importance of using protective gears (especially during farming activities) to protect them from being infected with schistosomiasis.*" (FGD 01, Healthcare workers, Itilima).

Third, they suggested that MDA of Praziquantel should be implemented in the communities too to cover other groups of people like adults who are currently not covered by school-based MDA programs.

"*If possible, mass drug administration should be implemented to cover all people. They should not be only for school children. I am not sure what criterion they use to implement it for school children only. It would be great if mass treatment would also be implemented among adults in the communities.*" (KII 10, Clinical Officer, Itilima).

## Discussion

This study, one of the few in sub-Saharan Africa, explored healthcare workers' KAP on FGS; healthcare facilies' capacity to manage FGS cases; and the challenges healthcare workers face when managing FGS cases. In general, although healthcare workers were aware of urogenital schistosomiasis, they had little to no knowledge of the symptoms, aetiology, and modes of transmission of FGS. They also had misconceptions regarding the transmission of FGS, particularly that it can be transmitted through sexual intercourse.

Urogenital schistosomiasis is a known public health problem reported by healthcare workers as important among children, adolescents, and adults. Healthcare workers were aware of the urinary symptoms of urogenital schistosomiasis. However, they had limited awareness that it could also affect the male and female reproductive tracts. Overall, healthcare workers' knowledge of urogenital schistosomiasis was limited and similar to what was reported in Ghana [28] including a misconception that schistosomiasis is sexually transmitted [29–31]. Importantly,

since healthcare workers regard urogenital schistosomiasis as sexually transmitted, they are more likely to refer women and adolescent girls presenting with FGS to the STI clinic or stigmatize them for perceived sexual promiscuity [28]. Adolescent girls reported stigmatization by healthcare workers as a key barrier for failing to access healthcare services [28]. Similarly, women with symptoms of FGS were reluctant to seek for treatment out of fear of being diagnosed with an STI that could lead to being stigmatized [32]. Taken together, these findings call for the need to train healthcare workers on FGS including how to managing its cases.

Misdiagnosing FGS or urogenital schistosomiasis as an STI is a common issue in endemic communities [29, 31]. Yet, studies have reported the association of FGS and HIV/AIDS [1, 33] as well as FGS and HPV/cervical cancer, gonorrhoea and syphilis [10]. Co-occurrences of these diseases with FGS have been widely reported among women during community screening [8]. Healthcare workers confuse the symptoms and clinical signs of FGS with those of STIs because they are similar [8], yet they do not consider FGS in their differential diagnoses and treatment plans unless the patient fails to respond to the initial therapy. Similarly, in Ghana, adolescent girls presenting with FGS symptoms were likely to be referred for STI treatment [28]. Because of these misconceptions and low knowledge of FGS, women with gynecologic symptoms usually use a combination of biomedical treatment (modern medicine), self-treatment/self-prescriptions and traditional medicine, with modern medicine being the last option. All these are crucial barriers that perpetuate undiagnosed and untreated FGS in many schistosomiasis endemic areas of sub-Saharan Africa [29–31].

The capacity of healthcare facilities to manage FGS cases were limited because of untrained healthcare workers, lack of knowledge of FGS, inadequate number of staff, and lack of medical equipment (e.g. miscroscopes and gynecologic specula) and supplies, including Praziquantel. Most of the healthcare workers reported that they had never been trained on FGS including the management of this disease. These major gaps in addressing FGS have also been reported in other parts of sub-Saharan Africa [3, 6]. Successful and sustainable integration of the FGS interventions with HIV and HPV/cervical prevention as part of a comprehensive SRH package requires availability of competent and skilled healthcare workers who can identify women and adolescent girls at risk, diagnose, and provide treatment [3, 6]. The creation of this comprehensive package will require in-service, competency-based training (moving beyond knowledge to "*Know-How*"). A competence-based training should encompass knowledge, attitudes, and skills to mitigate stigma. Lessons learned from HIV competency-based training to improve the management of FGS cases and mitigate stigma should be adapted to improve FGS competency-based training among healthcare workers [6]. In addition, stocking healthcare facilities with medical equipment and supplies (e.g. gynecologic specula, light source for examination, microscopes, slides, urine filter membraines and syringes and praziquantel) will further ensure sustainability of the management of FGS [6].

Stigmatization and lack of knowledge about FGS among women, adolescent girls, and healthcare workers are reported as some of the key barriers for health seeking among women and girls in endemic areas [28]. Community-based education was proposed as the best intervention to promote health seeking behavior among women and adolescent girls. This intervention should focus on creating awareness and filling the FGS knowledge gaps. As studies have shown, in order to be effective and successful, community-based education interventions focusing on changing health seeking behaviors can collaborate with existing community structures such as schools and religious institutions [34, 35]. Strategies that have been effectively used to create awareness include radio programs; information, education, and communication materials; community campaigns; and advocacy meetings with community leaders [34, 35]. The SASA! randomized trial conducted in Uganda provided evidence that using community

networks, identifying change agents, and propounding innovative media with stimulating messages led to improvement in behavior at both the individual and community levels [34].

Despite these interesting findings, our study had a few limitations. We acknowledge that our findings are based on healthcare workers reported responses and perceived experiences that we could not verify further. Second, because FGS was a new topic to most healthcare workers, some probes and explanations by the research assistants may have shaped the participants' narratives. We further acknowledge that while male genital schistosomiasis (MGS) is also a public health concern in the study settings [36], this study did not focus on it. Despite these limitations, we are convinced that our findings present a significant knowledge gap on FGS among healthcare workers, healthcare workers' little or no capacity to manage FGS cases, and what needs to be done to fix the situation.

## Conclusions and recommendations

Our findings demonstrated the high potential for education on FGS to fill knowledge gaps among healthcare workers. Our data suggest that in-service training should cover such topics as identification of women and adolescent girls at of risk; symptoms, aetiology, modes of transmission, and ways of preventing FGS; management of FGS (including the management of other STIs as part of the differential diagnosis); and mitigating stigma faced by women and girls suffering from FGS both at the healthcare facility settings and in the communities. As one participant emphasized when reflecting on the importance of educating healthcare workers on FGS, education would enable them to serve as "ambassadors" to educate the communities they serve. In addition, and as recommended by the study participants, healthcare facilities need to stocked with medical equipment and supplies to enable managing cases of FGS and other STIs. Filling these gaps will facilitate a smooth and sustainable integration of FGS interventions into a broader symptomatic and clinical management of STIs including HIV/AIDS and HPV/cervical cancer into a single but comprehensive SRH package.

## Supporting information

**S1 File. Focus group discussion guide health workers.**
(DOC)

**S2 File. Interview guide for key informants health workers.**
(DOC)

## Acknowledgments

The authors would like to thank the study participants for their time and responses without which this paper would not have been possible to write. We also thank the regional, district, shehia and village authorities where the study was conducted for granting the permission and providing every support to successfully conduct this study. Finally, we are grateful to Prof. Jennifer A. Downs of the Center for Global Health, Weill Cornell Medicine, New York, United States of America for her support in conceptualizing the project whose outputs including this paper.

## Author Contributions

**Conceptualization:** Humphrey D. Mazigo, Anna Samson.

**Data curation:** Humphrey D. Mazigo, Valencia J. Lambert, Agnes L. Kosia, Dunstan J. Matungwa.

**Formal analysis:** Humphrey D. Mazigo, Anna Samson, Dunstan J. Matungwa.

**Funding acquisition:** Humphrey D. Mazigo.

**Investigation:** Humphrey D. Mazigo, Anna Samson, Valencia J. Lambert, Dunstan J. Matungwa.

**Methodology:** Humphrey D. Mazigo, Anna Samson, Dunstan J. Matungwa.

**Project administration:** Humphrey D. Mazigo, Anna Samson.

**Supervision:** Humphrey D. Mazigo, Anna Samson, Valencia J. Lambert.

**Writing – original draft:** Humphrey D. Mazigo.

**Writing – review & editing:** Anna Samson, Agnes L. Kosia, Deogratias D. Ngoma, Rachel Murphy, Dunstan J. Matungwa.

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
